# Concept Learners for Few-Shot Learning

**Kaidi Cao**[*], **Maria Brbić**[*], **Jure Leskovec**
Department of Computer Science
Stanford University
{kaidicao, mbrbic, jure}@cs.stanford.edu

## Abstract

Developing algorithms that are able to generalize to a novel task given only a few labeled examples represents a fundamental challenge in closing the gap between machine- and human-level performance. The core of human cognition lies in the structured, reusable concepts that help us to rapidly adapt to new tasks and provide reasoning behind our decisions. However, existing meta-learning methods learn complex representations across prior labeled tasks without imposing any structure on the learned representations. Here we propose COMET, a meta-learning method that improves generalization ability by learning to learn along human-interpretable concept dimensions. Instead of learning a joint unstructured metric space, COMET learns mappings of high-level concepts into semi-structured metric spaces, and effectively combines the outputs of independent concept learners. We evaluate our model on few-shot tasks from diverse domains, including fine-grained image classification, document categorization and cell type annotation on a novel dataset from a biological domain developed in our work. COMET significantly outperforms strong meta-learning baselines, achieving 6–15% relative improvement on the most challenging 1-shot learning tasks, while unlike existing methods providing interpretations behind the model's predictions.

## 1 Introduction

Deep learning has reached human-level performance on domains with the abundance of large-scale labeled training data. However, learning on tasks with a small number of annotated examples is still an open challenge. Due to the lack of training data, models often overfit or are too simplistic to provide good generalization. On the contrary, humans can learn new tasks very quickly by drawing upon prior knowledge and experience. This ability to rapidly learn and adapt to new environments is a hallmark of human intelligence.

Few-shot learning (Miller et al., 2000; Fei-Fei et al., 2006; Koch et al., 2015) aims at addressing this fundamental challenge by designing algorithms that are able to generalize to new tasks given only a few labeled training examples. Meta-learning (Schmidhuber, 1987; Bengio et al., 1992) has recently made major advances in the field by explicitly optimizing the model's ability to generalize, or learning how to learn, from many related tasks (Snell et al., 2017; Vinyals et al., 2016; Ravi & Larochelle, 2017; Finn et al., 2017). Motivated by the way humans effectively use prior knowledge, meta-learning algorithms acquire prior knowledge over previous tasks so that new tasks can be efficiently learned from a small amount of data. However, recent works (Chen et al., 2019b; Raghu et al., 2020) show that simple baseline methods perform comparably to existing meta-learning methods, opening the question about which components are crucial for rapid adaptation and generalization.

Here, we argue that there is an important missing piece in this puzzle. Human knowledge is structured in the form of reusable concepts. For instance, when we learn to recognize new bird species we are already equipped with the critical concepts, such as wing, beak, and feather. We then focus on these specific concepts and combine them to identify a new species. While learning to recognize new species is challenging in the complex bird space, it becomes remarkably simpler once the reasoning is structured into familiar concepts. Moreover, such a structured way of cognition gives us the ability to provide reasoning behind our decisions, such as "ravens have thicker beaks than crows, with more

---

[*]The two first authors made equal contributions.

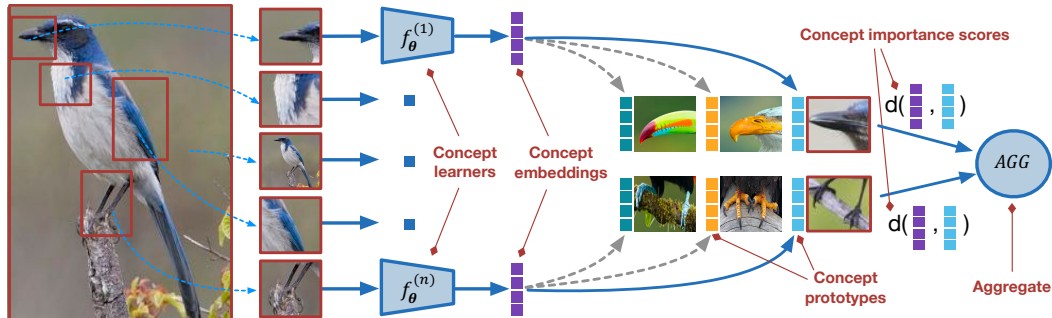

Figure 1: Along each concept dimension, COMET learns concept embeddings using independent concept learners and compares them to concept prototypes. COMET then effectively aggregates information across concept dimensions, assigning concept importance scores to each dimension.

of a curve to the end". We argue that this lack of structure is limiting the generalization ability of the current meta-learners. The importance of compositionality for few-shot learning was emphasized in (Lake et al., 2011; 2015) where hand-designed features of strokes were combined using Bayesian program learning.

Motivated by the structured form of human cognition, we propose COMET, a meta-learning method that discovers generalizable representations along human-interpretable concept dimensions. COMET learns a unique metric space for each *concept dimension* using concept-specific embedding functions, named *concept learners*, that are parameterized by deep neural networks. Along each high-level dimension, COMET defines *concept prototypes* that reflect class-level differences in the metric space of the underlying concept. To obtain final predictions, COMET effectively aggregates information from diverse concept learners and concept prototypes. Three key aspects lead to a strong generalization ability of our approach: (i) semi-structured representation learning, (ii) concept-specific metric spaces described with concept prototypes, and (iii) ensembling of many models. The latter assures that the combination of diverse and accurate concept learners improves the generalization ability of the base learner (Hansen & Salamon, 1990; Dvornik et al., 2019). Remarkably, the high-level universe of concepts that are used to guide our algorithm can be discovered in a fully unsupervised way, or we can use external knowledge bases to define concepts. In particular, we can get a large universe of noisy, incomplete and redundant concepts and COMET learns which subsets of those are important by assigning local and global concept importance scores. Unlike existing methods (Snell et al., 2017; Vinyals et al., 2016; Sung et al., 2018; Gidaris & Komodakis, 2018), COMET's predictions are interpretable—an advantage especially important in the few-shot learning setting, where predictions are based only on a handful of labeled examples making it hard to trust the model. As such, COMET is the first domain-agnostic interpretable meta-learning approach.

We demonstrate the effectiveness of our approach on tasks from extremely diverse domains, including fine-grained image classification in computer vision, document classification in natural language processing, and cell type annotation in biology. In the biological domain, we conduct the first systematic comparison of meta-learning algorithms. We develop a new meta-learning dataset and define a novel benchmark task to characterize single-cell transcriptome of all mouse organs (Consortium, 2018; 2020). Additionally, we consider the scenario in which concepts are not given in advance, and test COMET's performance with automatically extracted visual concepts. Our experimental results show that on all domains COMET significantly improves generalization ability, achieving 6–15% relative improvement over state-of-the-art methods in the most challenging 1-shot task. Furthermore, we demonstrate the ability of COMET to provide interpretations behind the model's predictions, and support our claim with quantitative and qualitative evaluations of the generated explanations.

## 2 PROPOSED METHOD

**Problem formulation**. In few-shot classification, we assume that we are given a labeled training set $\mathcal{D}_{tr}$, an unlabeled query set $\mathcal{D}_{qr}$, and a support set $\mathcal{S}$ consisting of a few labeled data points that share the label space with the query set. Label space between training and query set is disjoint, *i.e.,* $\{Y_{tr}\} \cap \{Y_{qr}\} = \emptyset$, where $\{Y_{tr}\}$ denotes label space of training set and $\{Y_{qr}\}$ denotes label space of

query set. Each labeled data point $(\mathbf{x}, y)$ consists of a $D$-dimensional feature vector $\mathbf{x} \in \mathbb{R}^D$ and a class label $y \in \{1, ..., K\}$. Given a training set of previously labeled tasks $\mathcal{D}_{tr}$ and the support set $\mathcal{S}$ of a few labeled data points on a novel task, the goal is to train a model that can generalize to the novel task and label the query set $\mathcal{D}_{qr}$.

## 2.1 PRELIMINARIES

**Episodic training**. To achieve successful generalization to a new task, training of meta-learning methods is usually performed using sampled mini-batches called episodes (Vinyals et al., 2016). Each episode is formed by first sampling classes from the training set, and then sampling data points labeled with these classes. The sampled data points are divided into disjoint sets of: (i) a support set consisting of a few labeled data points, and (ii) a query set consisting of data points whose labels are used to calculate a prediction error. Given the sampled support set, the model minimizes the loss on the sampled query set in each episode. The key idea behind this meta-learning training scheme is to improve generalization of the model by trying to mimic the low-data regime encountered during testing. Episodes with balanced training sets are usually referred to as "N-way, k-shot" episodes where $N$ indicates number of classes per episode ("way"), and $k$ indicates number of support points (labeled training examples) per class ("shot").

**Prototypical networks**. Our work is inspired by prototypical networks (Snell et al., 2017), a simple but highly effective metric-based meta-learning method. Prototypical networks learn a non-linear embedding function $f_{\boldsymbol{\theta}} : \mathbb{R}^D \to \mathbb{R}^M$ parameterized by a convolutional neural network. The main idea is to learn a function $f_{\boldsymbol{\theta}}$ such that in the $M$-dimensional embedding space data points cluster around a single prototype representation $\mathbf{p}_k \in \mathbb{R}^M$ for each class $k$. Class prototype $\mathbf{p}_k$ is computed as the mean vector of the support set labeled with the class $k$:

$$\mathbf{p}_k = \frac{1}{|\mathcal{S}_k|} \sum_{(\mathbf{x}_i, y_i) \in \mathcal{S}_k} f_{\boldsymbol{\theta}}(\mathbf{x}_i), \tag{1}$$

where $\mathcal{S}_k$ denotes the subset of the support set $\mathcal{S}$ belonging to the class $k$. Given a query data point $\mathbf{x}_q$, prototypical networks output distribution over classes using the softmax function:

$$p_{\boldsymbol{\theta}}(y = k|\mathbf{x}_q) = \frac{\exp(-d(f_{\boldsymbol{\theta}}(\mathbf{x}_q), \mathbf{p}_k))}{\sum_{k'} \exp(-d(f_{\boldsymbol{\theta}}(\mathbf{x}_q), \mathbf{p}_{k'}))}, \tag{2}$$

where $d : \mathbb{R}^M \to \mathbb{R}$ denotes the distance function. Query data point $\mathbf{x}_q$ is assigned to the class with the minimal distance between the class prototype and embedded query point.

## 2.2 META-LEARNING VIA CONCEPT LEARNERS

Our main assumption is that input dimensions can be separated into subsets of related dimensions corresponding to high-level, human-interpretable concepts that guide the training. Such sets of potentially overlapping, noisy and incomplete human-interpretable dimensions exists in many real-world scenarios. For instance, in computer vision concepts can be assigned to image segments; in natural language processing to semantically related words; whereas in biology we can use external knowledge bases and ontologies. In many problems, concepts are already available as a prior domain knowledge (Ashburner et al., 2000; Murzin et al., 1995; Wah et al., 2011; Mo et al., 2019; Miller et al., 2000), or can be automatically generated using existing techniques (Blei et al., 2003; Zhang et al., 2018; Jakab et al., 2018). Intuitively, concepts can be seen as part-based representations of the input and reflect the way humans reason about the world. Importantly, we do not assume these concepts are clean or complete. On the contrary, we show that even if there are thousands of concepts, which are noisy, incomplete, overlapping, or redundant, they still provide useful guidance to the meta-learning algorithm.

Formally, let $\mathcal{C} = \{\mathbf{c}^{(j)}\}_{j=1}^N$ denote a set of $N$ concepts given/extracted as a prior knowledge, where each concept $\mathbf{c}^{(j)} \in \{0, 1\}^D$ is a binary vector such that $c_i^{(j)} = 1$ if $i$-th dimension should be used to describe the $j$-th concept and $D$ denotes the dimensionality of the input. We do not impose any constraints on $\mathcal{C}$, meaning that the concepts can be disjoint or overlap. Instead of learning single mapping function $f_{\boldsymbol{\theta}} : \mathbb{R}^D \to \mathbb{R}^M$ across all dimensions, COMET separates original space into subspaces of predefined concepts and learns individual embedding functions $f_{\boldsymbol{\theta}}^{(j)} : \mathbb{R}^D \to \mathbb{R}^M$

for each concept $j$ (Figure 1). Concept embedding functions $f_{\boldsymbol{\theta}}^{(j)}$, named *concept learners*, are non-linear functions parametrized by a deep neural network. Each concept learner $j$ produces its own *concept prototypes* $\mathbf{p}_k^{(j)}$ for class $k$ computed as the average of concept embeddings of data points in the support set:

$$\mathbf{p}_k^{(j)} = \frac{1}{|\mathcal{S}_k|} \sum_{(\mathbf{x}_i, y_i) \in \mathcal{S}_k} f_{\boldsymbol{\theta}}^{(j)}(\mathbf{x}_i \circ \mathbf{c}^{(j)}), \tag{3}$$

where $\circ$ denotes Hadamard product. As a result, each class $k$ is represented with a set of $N$ concept prototypes $\{\mathbf{p}_k^{(j)}\}_{j=1}^N$.

Given a query data point $\mathbf{x}_q$, we compute its concept embeddings and estimate their distances to the concept prototypes of each class. We then aggregate the information across all concepts by taking sum over distances between concept embeddings and concept prototypes. Specifically, for each concept embedding $f_{\boldsymbol{\theta}}^{(j)}(\mathbf{x}_q \circ \mathbf{c}^{(j)})$ we compute its distance to concept prototype $\mathbf{p}_k^{(j)}$ of a given class $k$, and sum distances across all concepts to obtain a distribution over support classes. The probability of assigning query point $\mathbf{x}_q$ to $k$-th class is then given by:

$$p_{\boldsymbol{\theta}}(y = k|\mathbf{x}_q) = \frac{\exp(-\sum_j d(f_{\boldsymbol{\theta}}^{(j)}(\mathbf{x}_q \circ \mathbf{c}^{(j)}), \mathbf{p}_k^{(j)}))}{\sum_{k'} \exp(-\sum_j d(f_{\boldsymbol{\theta}}^{(j)}(\mathbf{x}_q \circ \mathbf{c}^{(j)}), \mathbf{p}_{k'}^{(j)}))}. \tag{4}$$

The loss is computed as the negative log-likelihood $L_{\boldsymbol{\theta}} = -\log p_{\boldsymbol{\theta}}(y = k|\mathbf{x}_q)$ of the true class, and COMET is trained by minimizing the loss on the query samples of training set in the episodic fashion (Snell et al., 2017; Vinyals et al., 2016). In equation (4), we use euclidean distance as the distance function. Experimentally, we find that it outperforms cosine distance (Appendix B), which agrees with the theory and experimental findings in (Snell et al., 2017). We note that in order for distances to be comparable, it is crucial to normalize neural network layers using batch normalization (Ioffe & Szegedy, 2015).

## 2.3 Interpretability

**Local and global concept importance scores**. In COMET, each class is represented with $N$ concept prototypes. Given a query data point $\mathbf{x}_q$, COMET assigns local concept importance scores by comparing concept embbeddings of the query to concept prototypes. Specifically, for a concept $j$ in a class $k$ the local importance score is obtained by inverted distance $d(f_{\boldsymbol{\theta}}^{(j)}(\mathbf{x}_q \circ \mathbf{c}^{(j)}), \mathbf{p}_k^{(j)})$. Higher importance score indicates higher contribution in classifying query point to the class $k$. Therefore, explanations for the query point $\mathbf{x}_q$ are given by local concept importance scores, and directly provide reasoning behind each prediction. To provide global explanations that can reveal important concepts for a set of query points of interest or an entire class, COMET computes average distance between concept prototype and concept embeddings of all query points of interest. Inverted average distance reflects global concept importance score and can be used to rank concepts, providing insights on important concepts across a set of examples.

**Discovering locally similar examples**. Given a fixed concept $j$, COMET can be used to rank data points based on the distance of their concept embeddings to the concept prototype $\mathbf{p}_k^{(j)}$ of class $k$. By ranking data points according to their similarity to the concept of interest, COMET can find examples that locally share similar patterns within the same class, or even across different classes. For instance, COMET can reveal examples that well reflect a concept prototype, or examples that are very distant from the concept prototype.

## 3 Experiments

### 3.1 Experimental setup

**Datasets**. We apply COMET to four datasets from three diverse domains: computer vision, natural language processing (NLP) and biology. In the computer vision domain, we consider fine-grained image classification tasks. We use bird classification CUB-200-2011 (Wah et al., 2011) and flower classification Flowers-102 (Nilsback & Zisserman, 2008) datasets, referred to as CUB and Flowers

hereafter. To define concepts, CUB provides part-based annotations, such as beak, wing, and tail of a bird. Parts were annotated by pixel location and visibility in each image. The total number of 15 parts/concepts is available; however concepts are incomplete and only a subset of them is present in an image. In case concept is not present, we rely on the prototypical concept to substitute for a missing concept. Based on the part coordinates, we create a surrounding bounding box with a fixed length to serve as the concept mask $\mathbf{c}^{(j)}$. On both CUB and Flowers datasets, we test automatic concept extraction. In NLP domain, we apply COMET to benchmark document classification dataset Reuters (Lewis et al., 2004) consisting of news articles. To define concepts, we use all hypernyms of a given word based on the WordNet hiearchy (Lewis et al., 2004). On all datasets, we include a concept that captures the whole input, corresponding to a binary mask of all ones.

In the biology domain, we introduce a new cross-organ cell type classification task (Brbić et al., 2020) together with a new dataset. We develop a novel single-cell transcriptomic dataset based on the Tabula Muris dataset (Consortium, 2018; 2020) that comprises $105,960$ cells of $124$ cell types collected across 23 organs of the mouse model organism. The features correspond to the gene expression profiles of cells. Out of the $23,341$ genes, we select $2,866$ genes with high standardized log dispersion given their mean. We define concepts using Gene Ontology (Ashburner et al., 2000; Consortium, 2019), a resource which characterizes gene functional roles in a hierarchically structured vocabulary. We select Gene Ontology terms at level 3 that have at least $64$ assigned genes, resulting in the total number of $190$ terms that define our concepts. We propose the evaluation protocol in which different organs are used for training, validation, and test splits. Therefore, a meta-learner needs to learn to generalize to unseen cell types across organs. This novel dataset along with the cross-organ evaluation splits is publicly available at `https://snap.stanford.edu/comet`. To our knowledge, this is the first meta-learning dataset from the biology domain.

**Baselines**. We compare COMET's performance to seven baselines, including FineTune/Baseline++ (Chen et al., 2019b), Matching Networks (MatchingNet) (Vinyals et al., 2016), Model Agnostic Meta-Learning (MAML) (Finn et al., 2017), Relation Networks (Sung et al., 2018), MetaOptNet (Lee et al., 2019), DeepEMD (Zhang et al., 2020) and Prototypical Networks (ProtoNet) (Snell et al., 2017). DeepEMD is only applicable to image datasets.

We provide more details on evaluation and implementation in Appendix A. Code is publicly available at `https://github.com/snap-stanford/comet`.

### 3.2 RESULTS

**Performance comparison**. We report results on CUB, Tabula Muris and Reuters datasets with concepts given as a prior domain knowledge in Table 1. COMET outperforms all baselines by a remarkably large margin on all datasets. Specifically, COMET achieves $9.5\%$ and $9.3\%$ average improvements over the best performing baseline in the 1-shot and 5-shot tasks across datasets. Notably, COMET improves the result of the ProtoNet baseline by $19$–$23\%$ in the 1-shot tasks across datasets. COMET's substiantial improvement are retained with the deeper Conv-6 backbone (Appendix C). To confirm that the improvements indeed come from concept learners and not from additional weights, we compare COMET to ensemble of prototypical networks, and further evaluate performance of COMET with shared weights across all concepts. Results shown in Table 2 demonstrate that COMET achieves significantly better performance than the ensemble of ProtoNets even when the weights across concepts are shared. Of note, COMET's performance is only slightly affected with shared weights across concepts. More experimental details are provided in Appendix D.

**Effect of number of concepts**. We systematically evaluate the effect of the number of concepts on COMET's performance on CUB and Tabula Muris datasets (Figure 2). In particular, we start from ProtoNet's result that can be seen as using a single concept in COMET that covers all dimensions of the input. We then gradually increase number of concepts and train and evaluate COMET with the selected number of concepts. For the CUB dataset, we add concepts based on their visibility frequency, whereas on the Tabula Muris we are not limited in the coverage of concepts so we randomly select them. The results demonstrate that on both domains COMET consistently improves performance when increasing the number of concepts. Strikingly, by adding just one most frequent concept corresponding to a bird's beak on top of the whole image concept, we improve ProtoNet's performance on CUB by $10\%$ and $5\%$ in 1-shot and 5-shot tasks, respectively. On the Tabula Muris, with just $8$ concepts COMET significantly outperforms all baselines and achieves $7\%$ and $17\%$

Table 1: Results on 1-shot and 5-shot classification on the CUB and Tabula Muris datasets. We report average accuracy and standard deviation over 600 randomly sampled episodes.

| | CUB | | Tabula Muris | | Reuters | |
|---|---|---|---|---|---|---|
| **Method** | **1-shot** | **5-shot** | **1-shot** | **5-shot** | **1-shot** | **5-shot** |
| **Finetune** | $61.4 \pm 1.0$ | $80.2 \pm 0.6$ | $65.3 \pm 1.0$ | $82.1 \pm 0.7$ | $48.2 \pm 0.7$ | $64.3 \pm 0.4$ |
| **MatchingNet** | $61.0 \pm 0.9$ | $75.9 \pm 0.6$ | $71.0 \pm 0.9$ | $82.4 \pm 0.7$ | $55.9 \pm 0.6$ | $70.9 \pm 0.4$ |
| **MAML** | $52.8 \pm 1.0$ | $74.4 \pm 0.8$ | $50.4 \pm 1.1$ | $57.4 \pm 1.1$ | $45.0 \pm 0.8$ | $60.5 \pm 0.4$ |
| **RelationNet** | $62.1 \pm 1.0$ | $78.6 \pm 0.7$ | $69.3 \pm 1.0$ | $80.1 \pm 0.8$ | $53.8 \pm 0.7$ | $68.3 \pm 0.3$ |
| **MetaOptNet** | $62.2 \pm 1.0$ | $79.6 \pm 0.6$ | $73.6 \pm 1.1$ | $85.4 \pm 0.9$ | $62.1 \pm 0.8$ | $77.8 \pm 0.4$ |
| **DeepEMD** | $64.0 \pm 1.0$ | $81.1 \pm 0.7$ | NA | NA | NA | NA |
| **ProtoNet** | $57.1 \pm 1.0$ | $76.1 \pm 0.7$ | $64.5 \pm 1.0$ | $82.5 \pm 0.7$ | $58.3 \pm 0.7$ | $75.1 \pm 0.4$ |
| **COMET** | $\mathbf{67.9 \pm 0.9}$ | $\mathbf{85.3 \pm 0.5}$ | $\mathbf{79.4 \pm 0.9}$ | $\mathbf{91.7 \pm 0.5}$ | $\mathbf{71.5 \pm 0.7}$ | $\mathbf{89.8 \pm 0.3}$ |

Table 2: Comparison to the ensemble of prototypical networks and COMET with shared weights across concepts. On the CUB dataset weights are always shared.

| | CUB | | Tabula Muris | | Reuters | |
|---|---|---|---|---|---|---|
| **Method** | **1-shot** | **5-shot** | **1-shot** | **5-shot** | **1-shot** | **5-shot** |
| **ProtoNetEns** | $64.0 \pm 0.8$ | $82.3 \pm 0.5$ | $67.2 \pm 0.8$ | $83.6 \pm 0.5$ | $62.4 \pm 0.7$ | $79.3 \pm 0.4$ |
| **COMET shared w** | $67.9 \pm 0.9$ | $85.3 \pm 0.5$ | $78.2 \pm 1.0$ | $91.0 \pm 0.5$ | $69.8 \pm 0.8$ | $88.6 \pm 0.3$ |
| **COMET** | $67.9 \pm 0.9$ | $85.3 \pm 0.5$ | $79.4 \pm 0.9$ | $91.7 \pm 0.5$ | $71.5 \pm 0.7$ | $89.8 \pm 0.3$ |

improvement over ProtoNet in 1-shot and 5-shot tasks, respectively. To demonstrate the robustness of our method to a huge set of overlapping concepts, we extend the number of concepts to 1500 by capturing all levels of the Gene Ontology hierarchy, therefore allowing many redundant relationships. Even in this scenario, COMET slightly improves the results compared to 190 concepts obtained from a single level. These results demonstrate that COMET outperforms other methods even when the number of concepts is small and annotations are incomplete, as well as with many overlapping and redundant concepts.

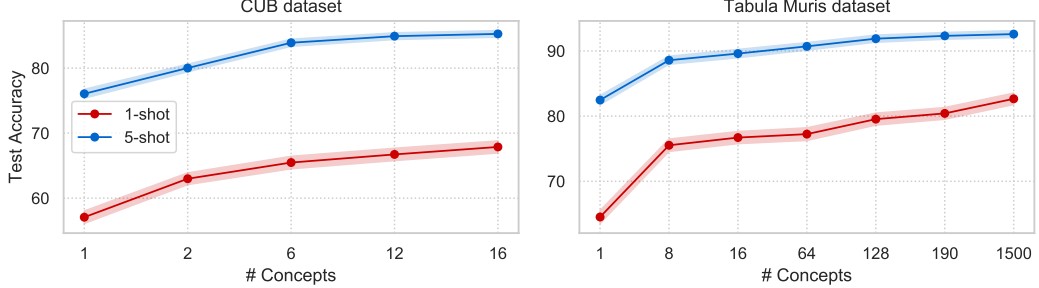

Figure 2: The effect of number of concepts on COMET's performance. COMET consistently improves performance when we gradually increase number of concept terms.

**Unsupervised concept annotation.** While COMET achieves remarkable results with human-validated concepts given as external knowledge, we next investigate COMET's performance on automatically inferred concepts. In addition to CUB dataset, we consider Flowers dataset for fine-grained image classification. To automatically extract visual concepts, we train the autoencoding framework for landmarks discovery proposed in (Zhang et al., 2018). The encoding module outputs landmark coordinates that we use as part coordinates. We generate a concept mask by creating a bounding box with a fixed length around landmark coordinates. Although extracted coordinates are often noisy and capture background (Appendix F), we find that COMET outperforms all baselines on both CUB and Flowers fine-grained classification datasets (Table 3). This analysis shows that the benefits of our method are expected even with noisy concepts extracted in a fully automated and unsupervised way.

To test unsupervised concept annotation on Tabula Muris and Reuters datasets, we randomly select subsets of features for concept definition. Since COMET is interpretable and can be used to find important concepts, we use validation set to select concepts with the highest importance scores. Even

in this case, COMET significantly outperforms all baselines, achieving only $2\%$ lower accuracy on the Tabula Muris dataset and $1\%$ on the Reuters dataset on both 1-shot and 5-shot tasks compared to human-defined concepts. This additionally confirms COMET's effectiveness with automatically extracted concepts. We provide more results in Appendix E .

Table 3: Results on 1-shot and 5-shot classification with automatically extracted concepts. We report average accuracy and standard deviation over 600 randomly sampled episodes. We show the average relative improvement of COMET over the best and ProtoNet baselines.

| Accuracy | CUB: 1-shot | CUB: 5-shot | Flowers: 1-shot | Flowers: 5-shot |
|---|---|---|---|---|
| **COMET** | $64.8 \pm 1.0$ | $82.0 \pm 0.5$ | $70.4 \pm 0.9$ | $86.7 \pm 0.6$ |
| **Improvement of COMET...** | | | | |
| **over best baseline** | 1.3% | 1.1% | 4.8% | 4.6% |
| **over ProtoNet** | 13.5% | 7.8% | 6.0% | 8.1% |

## 3.3 INTERPRETABILITY

We analyze the reasoning part of COMET by designing case studies aiming to answer the following questions: (i) Which concepts are the most important for a given query point (*i.e.*, local explanation)? Which concepts are the most important for a given class (*i.e.*, global explanation)?; (iii) Which examples share locally similar patterns?; (iv) Which examples reflect well concept prototype? We perform all analyses exclusively on classes from the novel task that are not seen during training.

**Concept importance**. Given a query point, COMET ranks concepts based on their importance scores, therefore identifying concepts highly relevant for the prediction of a single query point. We demonstrate examples of local explanations in Appendix G. To quantitatively evaluate global explanations that assign concept importance scores to the entire class, we derive ground truth explanations on the Tabula Muris dataset. Specifically, using the ground truth labels on the test set, we obtain a set of genes that are differentially expressed for each class (*i.e.*, cell type). We then find Gene Ontology terms that are significantly enriched (false discovery rate corrected $p$-value$< 0.1$) in the set of differentially expressed genes of a given class, and use those terms as ground-truth concepts. We consider only cell types that have at least two assigned terms. To obtain COMET's explanations, we rank global concept importance scores for each class and report the number of relevant terms that are successfully retrieved in top 20 concepts with the highest scores in the 5-shot setting (Figure 3 left). We find that COMET's importance scores agree extremely well with the ground truth annotations, achieving $0.71$ average recall@20 across all cell types. We further investigate global explanations on the CUB dataset by computing the frequency of the most relevant concepts across the species (Figure 3 right). Beak, belly and forehead turn out to be the most relevant features, supporting

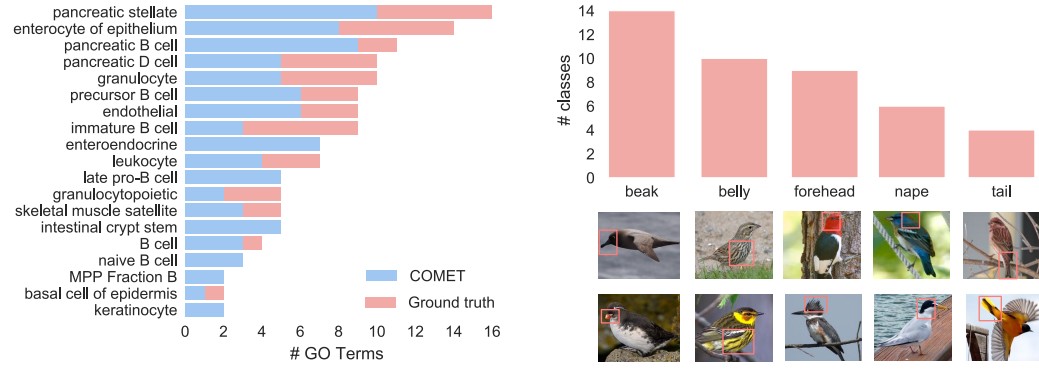

Figure 3: (Left) Quantitatively, on the Tabula Muris dataset COMET's global importance scores agree well with the ground truth important Gene Ontology terms estimated using differentially expressed genes. (Right) Qualitatively, on the CUB dataset importance scores correctly reflect the most relevant bird features.

common-sense intuition. For instance, 'beak' is selected as the most relevant concept for 'parakeet auklet' known for its nearly circular beak; 'belly' for 'cape may warbler' known for its tiger stripes on the belly; while 'belted kingfisher' indeed has characteristic 'forehead' with its shaggy crest on the top of the head. This confirms that COMET correctly identifies important class-level concepts.

**Locally similar patterns**. Given a fixed concept of interest, we apply COMET to sort images with respect to the distance of their concept embedding to the concept prototype (Figure 4). COMET finds images that locally resemble the prototypical image and well express concept prototype, correctly reflecting the underlying concept of interest. On the contrary, images sorted using the whole image as a concept often reflect background similarity and can not provide intuitive explanations. Furthermore, by finding most distant examples COMET can aid in identifying misannotated or non-visible concepts (Appendix H) which can be particularly useful when the concepts are automatically extracted. These analyses suggest that COMET can be used to discover, sort and visualize locally similar patterns, revealing insights on concept-based similarity across examples.

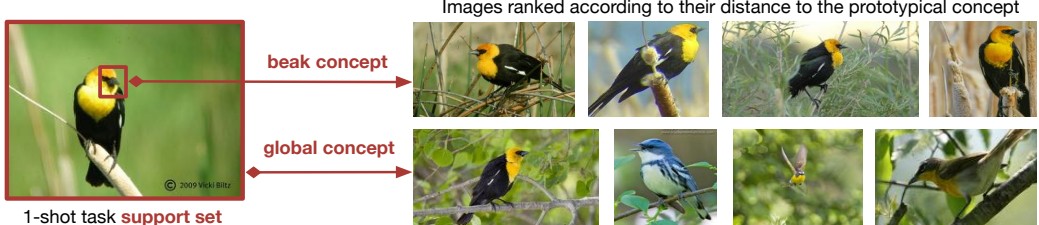

Figure 4: Top row shows images with beak concept embeddings most similar to the prototypical beak. Bottom row shows images ranked according the global concept that captures whole image. COMET correctly reflects local similarity in the underlying concept of interest, while global concept often reflects environmental similarity.

## 4 RELATED WORK

Our work draws motivation from a rich line of research on meta-learning, compositional representations, and concept-based interpretability.

**Meta-learning**. Recent meta-learning methods fall broadly into two categories. Optimization-based methods (Finn et al., 2017; Rusu et al., 2019; Nichol & Schulman, 2018; Grant et al., 2018; Antoniou et al., 2019) aim to learn a good initialization such that network can be fine-tuned to a target task within a few gradient steps. On the other hand, metric-based methods (Snell et al., 2017; Vinyals et al., 2016; Sung et al., 2018; Gidaris & Komodakis, 2018) learn a metric space shared across tasks such that in the new space target task can be solved using nearest neighbour or simple linear classifier. DeepEMD (Zhang et al., 2020) learns optimal distance between local image representations. Prototypical networks (Snell et al., 2017) learn a metric space such that data points cluster around a prototypical representation computed for each category as the mean of embedded labeled examples. It has remained one of the most competitive few-shot learning methods (Triantafillou et al., 2019), resulting in many follow-up works (Sung et al., 2018; Oreshkin et al., 2018; Ren et al., 2018; Liu et al., 2019; Xing et al., 2019). Two recent works (Hou et al., 2019; Zhu et al.) proposed to learn local discriminative features with attention mechanisms in image classification tasks. Our work builds upon prototypical networks and extends the approach by introducing concept-based prototypes. Prototypical networks were extended by learning mixture prototypes in (Allen et al., 2019); however prototypes in this work share the same metric space. In contrast, COMET defines human-interpretable concept-specific metric spaces where each prototype reflects class-level differences in the metric space of the corresponding concept.

**Compositionality**. The idea behind learning from a few examples using compositional representations originates from work on Bayesian probabilistic programs in which individual strokes were combined for the handwritten character recognition task (Lake et al., 2011; 2015). This approach was extended in (Wong & Yuille, 2015) by replacing hand designed features with symmetry axis as object descriptors. Although these early works effectively demonstrated that compositionality is a

key ingredient for adaptation in a low-data regime, it is unclear how to extend them to generalize beyond simple visual concepts. Recent work (Tokmakov et al., 2019) revived the idea and showed that deep compositional representations generalize better in few-shot image classification. However, this approach requires category-level attribute annotations that are impossible to get in domains not intuitive to humans, such as biology. Moreover, even in domains in which annotations can be collected, they require tedious manual effort. On the contrary, our approach is domain-agnostic and generates human-understandable interpretations in any domain.

**Interpretability**. There has been much progress on designing interpretable methods that estimate the importance of individual features (Selvaraju et al., 2016; Sundararajan et al., 2017; Smilkov et al., 2017; Ribeiro et al., 2016; Lundberg & Lee, 2017; Melis & Jaakkola, 2018). However, individual features are often not intuitive, or can even be misleading when interpreted by humans (Kim et al., 2018). To overcome this limitation, recent advances have been focused on designing methods that explain predictions using high-level human understandable concepts (Kim et al., 2018; Ghorbani et al., 2019). TCAV (Kim et al., 2018) defines concepts based on user-annotated set of examples in which the concept of interest appears. On the contrary, high-level concepts in our work are defined with a set of related dimensions. As such, they are already available in many domains, or can be obtained in an unsupervised manner. Once defined, they are transferable across problems that share feature space. As opposed to the methods that base their predictions on the posthoc analysis (Ribeiro et al., 2016; Lundberg & Lee, 2017; Melis & Jaakkola, 2018; Kim et al., 2018), COMET is designed as an inherently interpretable model and explains predictions by gaining insights from the reasoning process of the network. The closest to our work are prototypes-based explanation models (Li et al., 2018; Chen et al., 2019a). However, they require specialized convolutional architecture for feature extraction and are not applicable beyond image classification, or to a few-shot setting.

## 5 CONCLUSION

We introduced COMET, a novel metric-based meta-learning algorithm that learns to generalize along human-interpretable concept dimensions. We showed that COMET learns generalizable representations with incomplete, noisy, redundant, very few or a huge set of concept dimensions, selecting only important concepts for classification and providing reasoning behind the decisions. Our experimental results showed that COMET does not make a trade-off between interpretability and accuracy and significantly outperforms existing methods on tasks from diverse domains, including a novel benchmark dataset from the biology domain developed in our work.

## ACKNOWLEDGEMENTS

The authors thank Yusuf Roohani, Michihiro Yasunaga and Marinka Zitnik for their helpful comments. We gratefully acknowledge the support of DARPA under Nos. N660011924033 (MCS); ARO under Nos. W911NF-16-1-0342 (MURI), W911NF-16-1-0171 (DURIP); NSF under Nos. OAC-1835598 (CINES), OAC-1934578 (HDR), CCF-1918940 (Expeditions), IIS-2030477 (RAPID); Stanford Data Science Initiative, Wu Tsai Neurosciences Institute, Chan Zuckerberg Biohub, Amazon, JPMorgan Chase, Docomo, Hitachi, JD.com, KDDI, NVIDIA, Dell, Toshiba, and UnitedHealth Group. J. L. is a Chan Zuckerberg Biohub investigator.

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

## A    EXPERIMENTAL SETUP

**Evaluation**. We test all methods on the most broadly used 5-way classification setting. In each episode, we randomly sample 5 classes where each class contains $k$ examples as the support set in the $k$-shot classification task. We construct the query set to have 16 examples, where each unlabeled sample in the query set belongs to one of the classes in the support set. We choose the best model according to the validation accuracy, and then evaluate it on the test set with novel classes. We report the mean accuracy by randomly sampling 600 episodes in the fine-tuning or meta-testing stage.

On the CUB dataset, we followed the evaluation protocol in (Chen et al., 2019b) and split the dataset into 100 base, 50 validation, and 50 test classes in the exactly same split. On the Tabula Muris, we use 15 organs for training, 4 organs for validation, and 4 organs for test, resulting into 59 base, 47 validation, and 37 test classes corresponding to cell types. The 102 classes of Flowers dataset are split into 52, 25, 25 as the training, validation and testing set, respectively. As for Reuters dataset, we leave out 5 classes for validation and 5 for test.

**Implementation details**. On the CUB dataset, we use the widely adopted four-layer convolutional backbones Conv-4 with an input size of $84 \times 84$ (Snell et al., 2017). We perform standard data augmentation, including random crop, rotation, horizontal flipping and color jittering. We use the Adam optimizer (Kingma & Ba, 2014) with an initial learning rate of $10^{-3}$ and weight decay 0. We train the 5-shot tasks for $40,000$ episodes and 1-shot tasks for $60,000$ episodes (Chen et al., 2019b). To speed up training of COMET, we share the network parameters between concept learners. In particular, we first forward the entire image $\mathbf{x}_i$ into the convolutional network and get a spatial feature embedding $f_{\boldsymbol{\theta}}(\mathbf{x}_i)$, and then get the $j$-th concept embedding as $f_{\boldsymbol{\theta}}(\mathbf{x}_i) \circ \mathbf{c}^{(j)}$. Since convolutional filters only operate on pixels locally, in practice we get similar performance if we apply the mask at the beginning or at the end while significantly speeding up training time. In case the part is not annotatated (*i.e.*, visible), we use the prototypical concept corresponding to whole image to replace the missing concept. For the Tabula Muris dataset, we use a simple backbone network structure containing two fully-connected layers with batch normalization, ReLu activation and dropout. We use Adam optimizer (Kingma & Ba, 2014) with an initial learning rate of $10^{-3}$ and weight decay 0. We train the network for $1,000$ episodes. For MAML, RelationNet, MatchingNet, FineTune and ProtoNet, we use implementations from (Chen et al., 2019b). For MetaOptNet and DeepEMD we use implementations from the respective papers.

## B    ABLATION STUDY ON DISTANCE FUNCTION

We compare the effect of distance metric on the COMET's performance. We find that Euclidean distance consistently outperforms cosine distance in fine-grained image classification and cell type annotation tasks.

Table 4: The effect of distance metric on COMET's performance.

| Distance | CUB | | Tabula Muris | |
|---|---|---|---|---|
| | 1-shot | 5-shot | 1-shot | 5-shot |
| Cosine | $65.7 \pm 1.0$ | $82.2 \pm 0.6$ | $77.1 \pm 0.9$ | $90.1 \pm 0.6$ |
| **Euclidean** | $\mathbf{67.9 \pm 0.9}$ | $\mathbf{85.3 \pm 0.5}$ | $\mathbf{79.4 \pm 0.9}$ | $\mathbf{91.7 \pm 0.5}$ |

## C    ABLATION STUDY ON BACKBONE NETWORK

We compare performance of COMET to baselines methods using deeper Conv-6 backbone instead of Conv-4 backbone on the CUB dataset. We use part based annotations to define concepts. The results are reported in Table 5. COMET outperforms all baselines even with deeper backbone. Additionally, by adding just one most frequent concept corresponding to a bird's beak on top of the whole image concept, COMET improves ProtoNet's performance by $3.8\%$ on 1-shot task and $2.2\%$ on 5-shot task.

Table 5: Peformance using Conv-6 backbone on CUB and Flowers dataset. We report average accuracy and standard deviation over 600 randomly sampled episodes.

| Method | CUB | |
| --- | --- | --- |
| | 1-shot | 5-shot |
| Finetune | $66.0 \pm 0.9$ | $82.0 \pm 0.6$ |
| MatchingNet | $66.5 \pm 0.9$ | $77.9 \pm 0.7$ |
| MAML | $66.3 \pm 1.1$ | $78.8 \pm 0.7$ |
| RelationNet | $64.4 \pm 0.9$ | $80.2 \pm 0.6$ |
| MetaOptNet | $65.5 \pm 1.2$ | $83.0 \pm 0.8$ |
| DeepEMD | $66.8 \pm 0.9$ | $83.8 \pm 0.7$ |
| ProtoNet | $66.4 \pm 1.0$ | $82.0 \pm 0.6$ |
| COMET- 1 concept | $68.9 \pm 0.9$ | $83.8 \pm 0.6$ |
| COMET | $\mathbf{72.2 \pm 0.9}$ | $\mathbf{87.6 \pm 0.5}$ |

## D    ABLATION STUDY ON ENSEMBLE METHODS

We compare COMET to the ensemble of prototypical networks. We train ProtoNets in parallel and combine their outputs by majority voting as typically done in ensemble models. In particular, given a query point $\mathbf{x}_q$ and prototypes $\{\mathbf{p}_k^{(j)}\}_k$, the prototypical ensemble outputs probability distribution for each ProtoNet model $j$:

$$p_{\boldsymbol{\theta}}^{(j)}(y = k|\mathbf{x}_q) = \frac{\exp(-d(f_{\boldsymbol{\theta}}^{(j)}(\mathbf{x}_q), \mathbf{p}_k^{(j)}))}{\sum_{k'} \exp(-d(f_{\boldsymbol{\theta}}^{(j)}(\mathbf{x}_q), \mathbf{p}_{k'}^{(j)}))}. \tag{5}$$

On the CUB dataset, we use 5 ProtoNets. We use smaller number than the number of concepts because training an ensemble of a larger number of ProtoNets on CUB results in memory issues due to the unshared weights. On the Tabula Muris and Reuters datasets we use the same number of ProtoNets as the number of concepts, that is 190 on Tabula Muris and 126 on Reuters.

## E    UNSUPERVISED CONCEPT ANNOTATION: ADDITIONAL RESULTS

We evaluate COMET and baseline methods on the Flowers dataset for fine-grained image classification. We automatically extract concepts using unsupervised landmarks discovery approach (Zhang et al., 2018). Results in Table 6 show that COMET outperforms all baselines by a large margin.

Table 6: Results on 1-shot and 5-shot classification on the Flowers dataset. We report average accuracy and standard deviation over 600 randomly sampled episodes.

| Method | Flowers | |
| --- | --- | --- |
| | 1-shot | 5-shot |
| Finetune | $65.4 \pm 0.9$ | $81.9 \pm 0.7$ |
| MatchingNet | $66.0 \pm 0.9$ | $82.0 \pm 0.8$ |
| MAML | $63.2 \pm 1.1$ | $76.6 \pm 0.8$ |
| RelationNet | $66.4 \pm 0.9$ | $80.8 \pm 0.6$ |
| MetaOptNet | $64.8 \pm 1.0$ | $81.3 \pm 0.7$ |
| DeepEMD | $67.2 \pm 0.9$ | $82.9 \pm 0.7$ |
| ProtoNet | $64.4 \pm 1.0$ | $80.2 \pm 0.8$ |
| COMET | $\mathbf{70.4 \pm 0.9}$ | $\mathbf{86.7 \pm 0.6}$ |

On the Tabula Muris and Reuters datasets, we test COMET without any prior knowledge by defining concepts using selected random masks. In particular, we randomly select subsets of features as concepts and then use validation set to select the concepts with the highest importance scores as defined by COMET. We use same number of concepts used in Tabula Muris and Reuters datasets. Results are reported in Table 7.

## F    UNSUPERVISED CONCEPT ANNOTATION: LANDMARKS EXAMPLES

To assess the performance of COMET using automatically extracted visual concepts on the CUB dataset, we applied autoencoding framework for landmarks discovery proposed in (Zhang et al.,

Table 7: Results on 1-shot and 5-shot classification on Tabula Muris and Retuers dataset with selected random masks as concepts and human-defined concepts. We report average accuracy and standard deviation over 600 randomly sampled episodes.

| Method | Tabula Muris | | Reuters | |
|---|---|---|---|---|
| | 1-shot | 5-shot | 1-shot | 5-shot |
| with selected random masks | $77.2 \pm 1.0$ | $89.8 \pm 0.5$ | $70.1 \pm 0.9$ | $89.0 \pm 0.4$ |
| with prior knowledge | $\mathbf{79.4 \pm 0.9}$ | $\mathbf{91.7 \pm 0.5}$ | $\mathbf{71.5 \pm 0.7}$ | $\mathbf{89.8 \pm 0.3}$ |

2018). We use default parameters and implementation provided by the authors, and set the number of landmarks to 30. The encoding module provides coordinates of the estimated landmarks. To create concept mask, we create a bounding box around discovered landmarks. We train the autoencoder using same parameters as Zhang et al. (2018), and set the number of concepts to 30. Examples of extracted landmarks for 20 images from the CUB dataset are visualized in Figure 5.

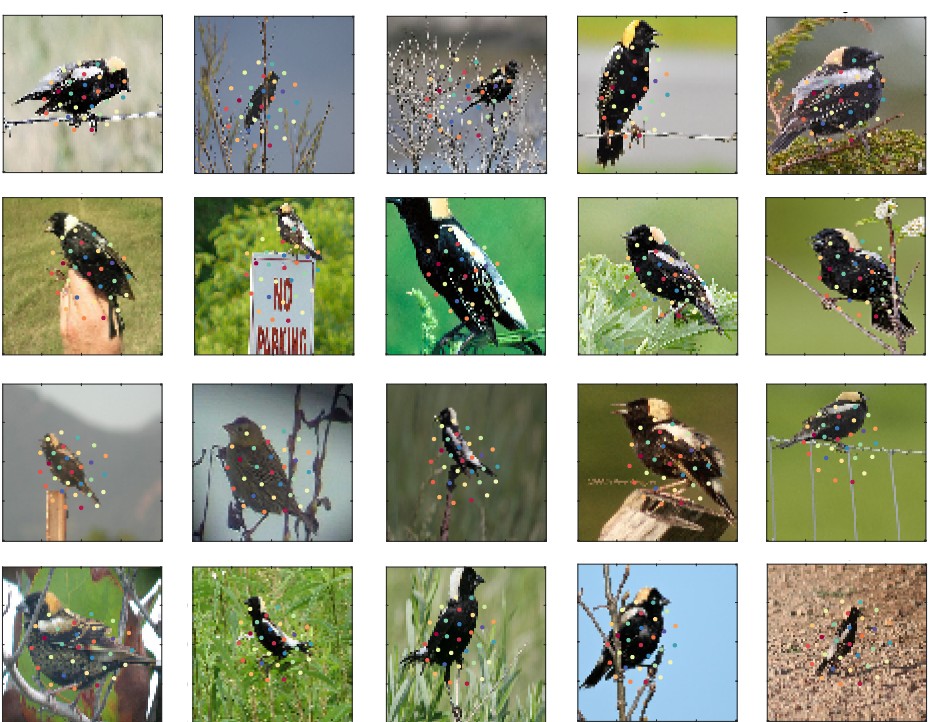

Figure 5: Examples of automatically extracted landmarks using (Zhang et al., 2018) on the CUB dataset.

## G  INTERPRETABILITY: LOCAL EXPLANATIONS

Here, we demonstrate COMET's local explanations on the CUB dataset. Given a single query data point, COMET assigns local concept importance scores to each concept based on the distance between concept embedding of the query data point to the prototypical concept. We then rank concepts according to their local concept importance scores. Figure 6 shows examples of ranked concepts. Importance scores assigned by COMET visually reflect well the most relevant bird features.

## H  INTERPRETABILITY: LOCAL SIMILARITY

Given fixed concept of interest, we apply COMET to sort images with respect to the distance of their concept embedding to the concept prototype. Figure 7 shows example of chipping sparrow images with the belly concept embedding most similar to the prototypical belly, and images with the belly

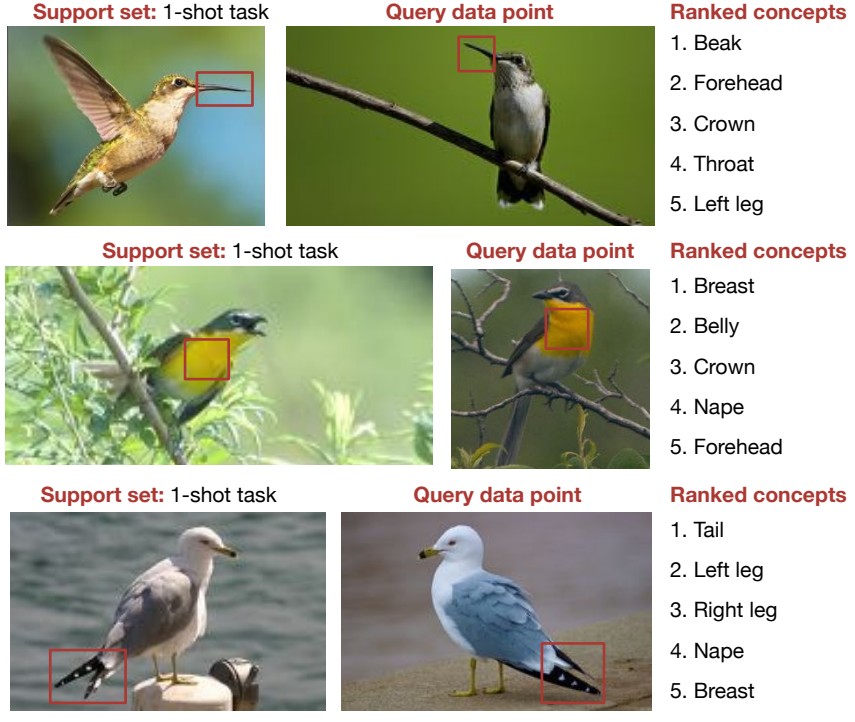

Figure 6: Examples of COMET's local explanations on the CUB dataset. Concepts are ranked according to the highest local concept similarity scores. Qualitatively, local importance scores correctly reflect the most relevant bird features.

concept embedding most distant to the prototypical belly. Most similar images indeed have clearly visible belly part and reflect prototypical belly well. On the contrary, most distant images have only small part of belly visible, indicating that COMET can be used to detect misannotated or non-visible concepts.

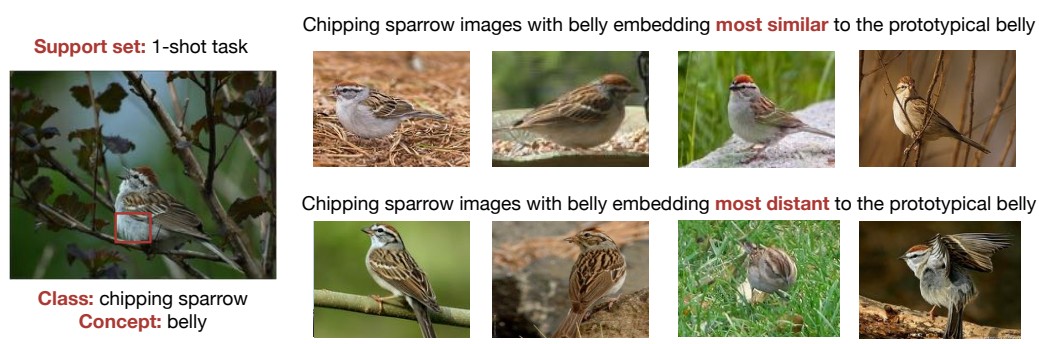

Figure 7: Images ranked according to the distance of their belly concept embedding to the belly concept prototype. Most similar images (top) and most distant images (bottom). Images closest to the prototype have clearly visible belly part that visually looks like prototypical belly of a chipping sparrow, whereas most distant images do not have belly part clearly visible.

