# OpenReview forum: "Concept Learners for Few-Shot Learning"
_ICLR.cc/2021/Conference — ICLR 2021 Poster_

### Official Review · AnonReviewer4 · 2020-10-28
**No obvious issues**

**Rating:** 7
**Confidence:** 4

**Review:**

Summary

This paper introduces potential use of intermediate structured representation of input space called “concepts” which are most likely human-interpretable. This intermediate space is then used for few-shot learning instead of using only the input space. This leads to better classification performance on the task, and it shows that injecting human-interpretable structured representation into task correlates with better performance (as one would hope). The paper uses datasets from different domains and shows improvement over approaches that don’t use the above defined “concepts”.


Strengths
- Paper is well-written
- Approach is tested on different dataset domains
- Experiments, ablation and qualitative results verify the claims of the paper


Weaknesses

- Figure 3 (Right)
    - In CUB you mentioned that not all 15 part-based annotations are available for every image (ref. Section 3.1 - Datasets). Does that mean there is a non-uniform distribution of part annotations per class? If this is the case, should it be taken into account for Figure 3 (Right)? It could be that beak is important for most of the classes simply because it was present more often that the other parts.

- Fine-grained classification only?
    - If I understood correctly, you need concepts (or part-based annotations) to be consistently present in all input images and all classes, because if they are not and the distribution of concepts is heavily skewed towards a subset of classes, then your approach won’t be effective. Does that mean that your approach is only effective for fine-grained classification datasets? Can mini-ImageNet or Omniglot be candidates for testing?


Minor concerns (suggestions, typos, etc.)
- Table 1
    - Are the methods reimplemented or you cited results from papers? Can you appropriately mention these facts and/or cite the papers where you picked numbers from in Table 1?
    - It was a bit surprising to see MAML underperforming in comparison to all the baselines (including Finetune)


Preliminary Rating and its justification

I see no obvious issues with the paper. I recommend accept.

---

> ### Author Response · Authors · 2020-11-18
> **Response to AnonReviewer4**
>
> We thank the reviewer for the very positive evaluation of our work and for recognizing our effort.
>
> *RE: It could be that beak is important for most of the classes simply because it was present more often that the other parts.*
>
> We thank the reviewer for the insightful comment. To investigate this question, we checked the frequency of top ranked concepts and found that they all have similar frequency (5206-5860). The lowest frequency (5206) has a “belly” concept; however it is ranked as the second most important concept.
>
> *RE: Does that mean that your approach is only effective for fine-grained classification datasets? Can mini-ImageNet or Omniglot be candidates for testing?*
>
> We thank the reviewer for this question and for understanding the paper correctly. In the computer vision domain, we consider fine-grained image classification since we can easily automatically extract concepts using landmark detection methods. Automatic concept extraction is more challenging on the mini-ImageNet and opens other research questions. In general, COMET can be applicable if at least one concept is shared between meta-train and meta-test datasets. Omniglot can be a candidate dataset, but since most methods achieve near-perfect performance, recent methods are not using it as a benchmark dataset anymore.
>
> *RE: Are the methods reimplemented or you cited results from papers? Can you appropriately mention these facts and/or cite the papers where you picked numbers from in Table 1?*
>
> For most baselines (MAML, ProtoNet, RelationNet, MatchingNet, FineTune) we built our codebase based on the implementation provided by Chen, et al. Closer Look at the Few-shot Classification (https://github.com/wyharveychen/CloserLookFewShot). For MetaOptNet and DeepEMD we used implementations from the respective papers. We rerun all methods in our codebase. We will make all code publicly available to the community, as well as contribute with the new benchmark datasets. We can also provide a private link to our codebase if needed.
>
> *RE: It was a bit surprising to see MAML underperforming in comparison to all the baselines (including Finetune)*
>
> We thank the reviewer for the careful observation. We did not change MAML implementation in any manner. On the CUB dataset, the only difference compared to the Closer Look at the Few-shot Classification paper by Cheng et al. is in the implementation of data augmentation. We used our own implementation but it is consistent across all experiments.

---

### Official Review · AnonReviewer2 · 2020-10-29
**Interesting idea for few-shot learning and some concerns on the experimental evaluation**

**Rating:** 6
**Confidence:** 3

**Review:**

The paper presents a knowledge-driven prototypical learning strategy for few-shot classification tasks. The main idea of this work is to introduce a set of concepts defined in the subspaces of inputs and represent each class as a group of concept prototypes for few-shot learning. Following the prototypical networks, the method first computes the concept embeddings of an input, and then takes the summation of the distances between those embeddings and their corresponding concept prototypes in each class to estimate the class probability.  The experiments validates the proposed methods on 4 benchmarks in three different domains, including vision, language and biology. For the biology task, the authors also develop a new benchmark on cross-organ cell type classification.

Strengths:
-  The idea of introducing concept knowledge into few-shot learning in a domain-agnostic manner seems interesting. While prior work has explored the compositionality for few-shot learning in vision domain, which shares the similar idea, the specific way that this paper introduces concepts as class representation is different.

-  The paper is well written and easy to follow.

- The experiments show the proposed method outperforms baseline prototypical network and other prior work on all four benchmarks across three domains.

Concerns:
-  Additional supervision from concepts: Using concept-based prototypes essentially introduces a form of part-level supervision, and therefore, it is not surprising to see the improvement in few-shot learning.  In addition, while such representations make sense for transferring prior knowledge on classes and provide interpretability, it is non-trivial to obtain such concepts in some domains. For example, it is difficult to learn the keypoint concepts for general object classes in vision domain.

- Eq. 4: The distance from an input to each class representation sums over individual distances from each concepts. It is unclear how such aggregation is able to handle noisy concepts and outliers. The summation would be dominated by concepts with large distances, which may not relevant to a specific input. More clarification would be helpful for this point.

- The experimental evaluation seems a bit lacking. For the vision domain, the paper only evaluated this method on the fine-grained classification tasks, which is uncommon in few-shot classification literature. It would be more convincing to show the results on the miniImageNet, which is a general object classification task, and see how the concept would be generalized to such cases.

- The method use multiple prototypes for each class, which has a higher model complexity than the standard prototypical network. Instead of using ensemble strategy (as shown in the suppl), what if the ablative study evaluates the setting in which the baseline learns multiple prototypes for each class?

- The effect of predefined concepts. Do the human-defined concepts play an critical role in the proposed method? What if the method uses randomly selected subsets of features for concept definition?

Updates: Thanks for the authors' response. The newly added experimental results address my concerns. I think this is an interesting work and recommend this paper to be accepted.

---

> ### Author Response · Authors · 2020-11-18
> **Response to AnonReviewer2**
>
> We thank the reviewer for the positive evaluation of our work. We are glad to hear that the reviewer appreciates the importance of developing domain-agnostic methods and recognizes validation of few-shot learning algorithms across diverse domains as one of the strengths of our work. While previous methods were focused on benchmarking on a few image datasets, we believe it is important to broaden the applicability of few-shot learning methods to other important domains. When applying few-shot learning in critical domains such as biology, interpretability of the method has an essential importance. Based on the reviewer’s insightful suggestions, we have conducted additional experiments that further validate our method.
>
> *RE: While such representations make sense for transferring prior knowledge on classes and provide interpretability, it is non-trivial to obtain such concepts in some domains. For example, it is difficult to learn the keypoint concepts for general object classes in vision domain.*
>
> We thank the reviewer for this important question and we agree that it is difficult to learn the keypoint concepts for general object classes in the computer vision domain. In our work we focus on fine-grained image classification where the concepts can be easily automatically extracted using landmark detection methods. In other domains such as biology, medicine and NLP the concepts are very easily obtainable. For example, WordNet can be used for general NLP tasks, while in medical and biological applications knowledge is often represented using ontologies and hierarchical structures such as GeneOntology, CellOntology, or phylogenetic tree. We note that the relationships between features in COMET could be also given as a graph and features could be grouped based on different subgraphs, for example gene regulatory or protein-protein interaction networks.
>
> *RE: It is unclear how such aggregation is able to handle noisy concepts and outliers.*
>
> The reviewer is correct that the summation would be dominated by the nosy concept. However, we normalize summation across classes meaning that in such case distance would be large across all the classes (denominator in our equation (4)), so other non-noisy concepts would make a difference when assigning a data point to a particular class. This is also experimentally validated in our experiments with automatic concept extraction. Landmarks shown in Appendix F are often noisy and capture background; however, COMET still improves performance over baselines. Additionally, the Tabula Muris experiment in Figure 2 (right) shows that the method is robust to the extremely large number of redundant concepts.
>
> *RE: For the vision domain, it would be more convincing to show the results on the miniImageNet, which is a general object classification task.*
>
> We thank the reviewer for the question. In our work we focus on fine-grained image classification in the computer vision domain. This allows us to easily automatically extract concepts. It is challenging to automatically extract keypoints for general object classes and this opens other research questions that are out of scope of this work. We leave this for the future work.
>
> *RE: What if the ablative study evaluates the setting in which the baseline learns multiple prototypes for each class?*
>
> We thank the reviewer for the question. We have now tested the baseline with multiple prototypes for each class proposed by the reviewer. We found that this baseline consistently achieves worse performance than the ProtoNets ensemble baseline reported in the paper with the same number of prototypes/ProtoNets. Specifically, on the Tabula Muris dataset, it achieves 2.1% lower accuracy on 1-shot task and 0.6% lower accuracy on 5-shot task. On the Reuters dataset, it achieves 2.8% lower accuracy on 1-shot task and 2.4% lower accuracy on 5-shot task. Consequently, COMET significantly outperforms this baseline.
>
> *RE: Do the human-defined concepts play a critical role in the proposed method? What if the method uses randomly selected subsets of features for concept definition?*
>
> We thank the reviewer for this very interesting question. In addition to the unsupervised concept learning on image datasets, we have also tested COMET on the Tabula Muris and Reuters datasets without any prior knowledge. Since COMET is interpretable and can be used to find important concepts, we randomly select feature subsets and use validation set to select those subsets with the highest importance scores. COMET significantly outperforms all baselines, achieving only 2% lower accuracy on Tabula Muris and 1% on Reuters on both 1-shot and 5-shot tasks compared to human-defined concepts. This additionally confirms COMET’s effectiveness with automatically extracted concepts. We have included these results in Table 7 in Appendix E and we thank the reviewer for suggesting this interesting experiment which helped to improve the paper.

---

### Official Review · AnonReviewer3 · 2020-10-30
**The paper need more clarification.**

**Rating:** 5
**Confidence:** 4

**Review:**

This paper proposes concept learners to effectively combines the outputs of independent concept learners. The model is evaluated on several datasets from different domains.

First of all, why the authors define it as generalizable few-shot learning, the settings targeted in this paper seem to do no different from traditional few-shot learning. Why is it called generalizable few-shot learning?

The other two main concerns are:

The idea of learning to attend different segments of an image or learning to the segment has been proposed in previous literature [a,b,c,d]. Even if they are not specifically targeted on a few-shot image classification, the proposed concept learners are still pretty similar to previous works and are not specifically designed for few-shot image classification tasks. Thus, I believe the novelty is somewhat limited for this submission.

In experiments, even if the authors choose three datasets for comparisons. I am more interested in results on standard benchmarks, such as miniImageNet, tieredImageNet. The results on the current datasets are not a convincing performance in my point of view. It is expected to see experiments on more standard and large-scaled datasets.

A minor point I am curious about is that by simple data augmentation method: crop, is it possible that multiple random cropping can generate different concepts and achieve similar effects by simply cropping multiple times on one image?


[a] Linsley, D., Shiebler, D., Eberhardt, S. and Serre, T., 2018. Learning what and where to attend. arXiv preprint arXiv:1805.08819.

[b] B. Zhou, A. Khosla, A. Lapedriza, A. Oliva, and A. Torralba. Learning Deep Features for Discriminative Localization. CVPR'16 (arXiv:1512.04150, 2015).

[c] Zhu Y, Liu C, Jiang S. Multi-attention meta learning for few-shot fine-grained image recognition[C]//Twenty-Ninth International Joint Conference on Artificial Intelligence and Seventeenth Pacific Rim International Conference on Artificial Intelligence. 2020: 1090-1096.

[d] Hou R, Chang H, Bingpeng M A, et al. Cross attention network for few-shot classification[C]//Advances in Neural Information Processing Systems. 2019: 4003-4014.

---

> ### Author Response · Authors · 2020-11-18
> **Response to AnonReviewer3**
>
> We thank the reviewer for the valuable feedback.
>
> *RE: Why is it called generalizable few-shot learning?*
>
> We thank the reviewer for this question. Our work is motivated by the way human knowledge is structured in the form of reusable concepts that help us to generalize to unseen tasks. The goal in our work is to show that introducing such structure in meta-learners improves generalization ability, as supported by our experiments. The setting is not different from other few-shot learning methods, but our key idea of transferable concepts results in significantly improved generalization ability over the existing few-shot learning methods. We agree with the reviewer that the proposed title may be misleading. We removed the word “generalizable” from the title as suggested by the reviewer.
>
> *RE: The idea of learning to attend different segments of an image or learning to the segment has been proposed in previous literature [a,b,c,d]. Thus, I believe the novelty is somewhat limited for this submission.*
>
> We thank the reviewer for the references. However, the referenced works are introducing attention-mechanism into convolutional architecture in order to learn to attend over different image regions. This differs greatly from the approach proposed in our work in the number of aspects. First, we are not introducing attention mechanisms in the network, but relying on the high-level human-understandable concepts. Next, the key idea in our work is that combining diverse and independent concept learners defined in concept-specific metric spaces helps us to generalize better in the few-shot scenario. In contrast, references [a-d] learn only one metric space with more discriminative features obtained with the attention mechanism. Finally, all these works are focused on learning to attend on image classification tasks; while our approach is domain-agnostic and can be effectively applied in diverse domains providing also interpretations behind predictions, as demonstrated in our experiments. In response to the reviewer’s feedback, we have included [c,d] in the related work.
>
> *RE: In experiments, even if the authors choose three datasets for comparisons. I am more interested in results on standard benchmarks, such as miniImageNet, tieredImageNet. The results on the current datasets are not a convincing performance in my point of view.*
>
> We thank the reviewer for the question and we would just like to correct that we evaluate performance on four datasets in three domains. In the computer vision domain, we evaluate performance on CUB and Flowers datasets which are standard benchmarks for fine-grained image classification tasks. Reference [c] suggested by the reviewer considers a few-shot classification only in the fine-grained image recognition task. As emphasized in [c], fine-grained image recognition is more difficult than general image recognition due to the subtle and local differences. Fine-grained image classification allows us to automatically extract concepts easily. On general image classification task automatic concept extraction opens other research questions that are out of scope of this work. We would like to emphasize that unlike previous methods which were focused on benchmarking on image datasets, our method is domain-agnostic and broadens the applicability of few-shot learning to other domains. Reuters is a standard benchmark for text classification, while in the biology domain we contribute to the community with the new dataset and conduct the first systematic evaluation of meta-learning methods.
>
> *RE:  By simple data augmentation method: crop, is it possible that multiple random cropping can generate different concepts and achieve similar effects by simply cropping multiple times on one image?*
>
> We thank the reviewer for the question, but this is not possible. Random crops would not be transferable across images. Concepts are defined as meaningful object structures that appear across domains and tasks.

---

### Official Review · AnonReviewer5 · 2020-11-06
**Review for Concept Learners for Generalizable Few-Shot Learning**

**Rating:** 6
**Confidence:** 4

**Review:**

**Overview**
Inspired by DeepEMD's obersvation that compositional representations generalize better for few-shot image classification, this paper (COMET) introduces "Concepts Embeddings" components to the Prototypical Networks. "Concepts Embeddings" are part-based representations and are learnt by a set of independent networks $ \\{ f_{\theta_i} \\} $ (can also share weights).

For each novel class, its $N$ concept prototypes are computed using the support set, and a query example is classified by measuring its distances to each novel class's $N$ concept prototypes.

Evaluations and experiments are carried out on 3 datasets: CUB, Tabula Muris and Reuters. The proposed COMET approach obtains favorable performances on all 3 datasets.

**Pros**
This work explore ways to learn compositional representations in a semi-supervised fashion.  It takes Prototypical Networks as baseline and adds a novel "Concept Embedding" component.  The authors show the proposed approach can significantly improve ProtoNet baseline on 3 datasets. It also outperforms the recent approach DeepEMD. The following observation on the CUB dataset is very interesting: "Strikingly, by adding just one most frequent concept corresponding to a bird’s beak on top of the whole image concept, we improve ProtoNet’s performance on CUB by 10% and 5% in 1-shot and 5-shot tasks, respectively."

**Cons**
Using multiple prototypes per class has been explored in "Infinite Mixture Prototypes for Few-shot Learning" (ICML'19), maybe the authors should mention it and highlight differences.

It's unclear where does $C$ comes from by just reading section 2.2.  How to compute $C$ is later elaborated in experiments section 3.1. Essentially $C$ are hand-crafted for each dataset. Maybe the authors should hint this in section 2.2.

The math symbols are a bit confusing in section 2.2. For instance, inconsistent notation $f_{\theta}^{(j)}$ in section 2.2 and $f_{\theta_j}$ in Figure 1, also $D$'s definition is not given (is it equal to the number of pixels $\times$ number of input channel?).

I appreciate the authors provinding many interesting ablation studies, but the main evaluation (Table 1) misses 2 important image few-shot classification dataset/benchmarks: miniImageNet, and MetaDataset.  How does the COMET with "Unsupervised concept annotation" performs on the above two datasets?

When compared with ensemble, why specifically 5 ProtoNets? How many concepts are used in COMET and COMET (shared weight)? This ablation should be included in main paper, as network capacity can improve accuracy significantly, see: "A Closer Look at Few-shot Classification"(ICLR'19).


**Reason for the decision**
This paper explores using compositional representations in Prototypical Networks. The authors introduce a novel concept embeddings idea, and show that it can significantly improve ProtoNet baseline on 3 datasets. However using multiple prototypes per class is not a new idea and the authors should highlight the differences with prior works (what's the strength of this particular formulation?). For the evaluation, I highly suggest the authors to include standard mini-ImageNet and Meta-Dataset FSL image classification benchmarks and report performances on them.

I find the following observation on the CUB dataset  very interesting: "Strikingly, by adding just one most frequent concept corresponding to a bird’s beak on top of the whole image concept, we improve ProtoNet’s performance on CUB by 10% and 5% in 1-shot and 5-shot tasks, respectively." That means human-validate concepts are complimentary to the representations learnt by a 4-layer network, but how about using a deep/stronger backbone, would it narrow the margin? I suggest the authors add this ablation study: How much does adding a concept help w.r.t to the depth of the backbone.

---

> ### Author Response · Authors · 2020-11-18
> **Response to AnonReviewer5**
>
> We thank the reviewer for a very valuable feedback that helped us to improve the paper. We have conducted additional experiments and included results of the new experiments as well as other reviewer’s suggestions in the revised paper.
>
> *RE: Using multiple prototypes per class has been explored in "Infinite Mixture Prototypes for Few-shot Learning" (ICML'19), maybe the authors should mention it and highlight differences.*
>
> We thank the reviewer for the relevant reference. Infinite Mixture Prototypes (IMP) learns multiple prototypes per class; however, these prototypes are defined in the same metric space and obtained by clustering the support set into class-wise means. In contrast, COMET defines concept-specific metric spaces and each prototype reflects class-level differences in the metric space of the corresponding concept. The strength of COMET’s particular formulation lies in combining diverse concept-specific embedding functions which improves generalization ability. Additionally, since prototypes are defined along human-interpretable dimensions, COMET’s predictions are interpretable in contrast to IMP.  In response to the reviewer’s feedback, we have included IMP in the related work and highlighted differences.
>
> *RE: It's unclear where does C comes from by just reading section 2.2. Essentially C are hand-crafted for each dataset. Maybe the authors should hint this in section 2.2. *
>
> The reviewer is correct that we assume that concepts C are given as a prior knowledge. We have revised the text as suggested by the reviewer.
>
> *RE: For instance, inconsistent notation fθ(j)in section 2.2 and fθj in Figure 1, also D's definition is not given (is it equal to the number of pixels ×number of input channel?).*
>
> We thank the reviewer for this great observation and checking our paper carefully. We have revised the notation in the Figure 1 to be consistent with the notation in section 2.2. Additionally, we have defined D as the number of dimensions. In the image-classification task, D corresponds to the number of pixels ×number of input channels.
>
> *RE:  How does the COMET with "Unsupervised concept annotation" performs on the above miniImageNet, and MetaDataset datasets?*
>
> We thank the reviewer for the question. In this work we focus on the fine-grained image classification tasks. This allows us to easily automatically extract concepts. Automatic concept extraction on datasets such as miniImageNet and MetaDataset opens other research questions that are out of scope of this work. We leave this for the future work.
>
> *RE: Why specifically 5 ProtoNets? How many concepts are used in COMET and COMET (shared weight)? This ablation should be included in main paper.*
>
> We thank the reviewer for this insightful comment. On the CUB dataset, we use 15 concepts and we always share weights between concept learners. The reason behind 5 ProtoNets is that training an ensemble of a larger number of ProtoNets on CUB results in memory issues due to the unshared weights. On the Tabula Muris dataset we use 190 concepts, and on Reuters 126 concepts. When comparing performance on these datasets, we use the same number of ProtoNets as the number of concepts, that is 190 and 126 ProtoNets. We have moved this ablation study to the main paper as suggested by the reviewer.
>
> *RE: How about using a deep/stronger backbone, would it narrow the margin? I suggest the authors add this ablation study: How much does adding a concept help w.r.t to the depth of the backbone.*
>
> We thank the reviewer for this important question. Based on the reviewer’s feedback, we have tested COMET as well as other baselines with the Conv-6 instead of Conv-4 backbone. With Conv-6 backbone, COMET achieves 8% improvement over DeepEMD as the best baseline method and 8.7% improvement over ProtoNet on the 1-shot task.  On the 5-shot task, COMET achieves 4.5% and 6.8% improvements over DeepEMD and ProtoNet, respectively. These results confirm effectiveness of COMET even with the deeper backbone network. Additionally, we have tested the improvements of adding only one concept in COMET with the Conv-6 backbone. Adding just the “beak” concept improves ProtoNet’s performance by 3.8% on 1-shot task and 2.2% on 5-shot task. We have included these results in Table 5 in Appendix C titled Ablation study on backbone network. We thank the reviewer for suggesting this experiment.

---

### Decision · Program_Chairs · 2021-01-07
**Final Decision**

**Decision:**

Accept (Poster)

**Comment:**

The paper introduces "Concept Embeddings"  to  Prototypical Network, which are part-based representations and are learnt by a set of independent networks (which can share weights).  The method first computes the concept embeddings of an input, and then takes the summation of the distances between those concept embeddings and their corresponding concept prototypes in each class to estimate the class probability. The experiments validates the proposed methods on 4 benchmarks in three different domains, including vision, language and biology. For the biology task, the authors also develop a new benchmark on cross-organ cell type classification.  The key novel idea of transferable concepts results in significantly improved generalization ability over the existing few-shot learning methods.

Although some reviewers raised concerns about not using other few-shot image classification datasets such as MiniImageNet these are not appropriate benchmarks, as the method requires the “part-based concepts” to reasonably span the space of all images which is a characteristic of fine-grained image classification problem.   Although this does limit the scope of the method, the fact that it is applicable for multiple tasks is a strong counteragument to the claim that it is too limited, so overall I disagree with the assessment of one reviewer that the choice of benchmarks is insufficient.